# A larger augmented-reality field of view improves interaction with nearby holographic objects

**Eva M. Hoogendoorn**\*, **Daphne J. Geerse, Jip Helsloot, Bert Coolen, John F. Stins, Melvyn Roerdink**\*

Department of Human Movement Sciences, Faculty of Behavioural and Movement Sciences, Amsterdam Movement Sciences, Vrije Universiteit Amsterdam, Amsterdam, Netherlands

\* e.m.hoogendoorn@vu.nl (EMH); m.roerdink@vu.nl (MR)

**Data Availability Statement:** All relevant data are within the manuscript and its Supporting Information files.

## Abstract

Augmented-reality (AR) applications have shown potential for assisting and modulating gait in health-related fields, like AR cueing of foot-placement locations in people with Parkinson's disease. However, the size of the AR field of view (AR-FOV), which is smaller than one's own FOV, might affect interaction with nearby floor-based holographic objects. The study's primary objective was to evaluate the effect of AR-FOV size on the required head orientations for viewing and interacting with real-world and holographic floor-based objects during standstill and walking conditions. Secondary, we evaluated the effect of AR-FOV size on gait speed when interacting with real-world and holographic objects. Sixteen healthy middle-aged adults participated in two experiments wearing HoloLens 1 and 2 AR headsets that differ in AR-FOV size. To confirm participants' perceived differences in AR-FOV size, we examined the head orientations required for viewing nearby and far objects from a standstill position (Experiment 1). In Experiment 2, we examined the effect of AR-FOV size on head orientations and gait speeds for negotiating 2D and 3D objects during walking. Less downward head orientation was required for looking at nearby holographic objects with HoloLens 2 than with HoloLens 1, as expected given differences in perceived AR-FOV size (Experiment 1). In Experiment 2, a greater downward head orientation was observed for interacting with holographic objects compared to real-world objects, but again less so for HoloLens 2 than HoloLens 1 along the line of progression. Participants walked slightly but significantly slower when interacting with holographic objects compared to real-world objects, without any differences between the HoloLenses. To conclude, the increased size of the AR-FOV did not affect gait speed, but resulted in more real-world-like head orientations for seeing and picking up task-relevant information when interacting with floor-based holographic objects, improving the potential efficacy of AR cueing applications.

**Funding:** This publication was funded by 1) project 'Holocue: Assisting gait in Parkinson's disease with intelligent mixed-reality cueing' (with project number 19357) which is financed by the Demonstrator programme of the Dutch Research Council (NWO)(https://www.nwo.nl/). The funders had no role in study design, data collection and analysis, decision to publish, or preparation of the manuscript and 2) EMIL project financial support to third parties, which is funded by the European Union. Views and opinions expressed are, however, those of the author(s) only and do not necessarily reflect those of the European Union. Neither the European Union nor the granting authority can be held responsible for them. MR and DJG received both grants.

**Competing interests:** MR is a scientific advisor with share options for Strolll Ltd., a digital therapeutics company building AR software for physical rehabilitation, for one day a week ancillary to his main position as associate professor Technology in Motion at Vrije Universiteit Amsterdam. The other authors declare no conflicts of interest. This does not alter our adherence to PLOS ONE policies on sharing data and materials. There are no patents, products in development or marketed products associated with this research to declare.

## Introduction

Leading augmented-reality (AR) glasses, like Microsoft HoloLens and Magic Leap, are progressing towards consumer readiness, with profound improvements in wearer comfort, form factor, data quality, and AR field of view (AR-FOV). One of the strengths of AR compared to virtual reality is that the visible real world can be augmented with environment-aware digital content, facilitating direct bodily interaction with digital objects. AR has therefore gained interest in health-related fields like rehabilitation and physical therapy given its potential efficacy for gait assistance [1, 2], gait modulation [3–5] gait-and-balance assessment [6–10] and gait-and-balance training. In these fields, AR allows the user to experience the real world augmented with (interactive) holographic floor-based objects like 2D AR cues to step onto or 3D AR obstacles to step over, an action-relevant gait-guidance technique which could improve gait parameters like stride length, step length and walking velocity in gait-impaired individuals [11–15].

In people with Parkinson's disease (PD), for example, AR has been explored for providing external visual cues [1, 2]. External cueing (i.e., the use of external temporal or spatial stimuli) is a well-established strategy for alleviating freezing of gait (FOG) and facilitating gait in people with PD [16]. Visual cues were originally applied with 2D stripes taped to the ground as targets for foot placement, but recent insights suggest that 3D cues, like an obstacle to cross, could be more effective than 2D cues for modifying gait and alleviating FOG [17, 18]. By using wearable AR glasses, such 2D or 3D holographic objects could be presented anywhere, anytime instead of being location-bound as with real-world 2D or 3D cues, thereby potentially broadening their applicability to everyday-life situations.

Geerse et al. examined the effect of 2D and 3D AR visual cueing using Microsoft HoloLens 1 (see Fig 1A) on FOG in the home environment of people with PD [1]. Objective and subjective immediate improvements in the number and duration of FOG episodes were found for participants with long and/or many FOG episodes [1]. Participants reported that the wearer comfort and the limited AR-FOV of HoloLens 1 ($30° \times 17°$ [19]) needed improvement [1], as they experienced the holographic objects as if they were watching them through a relatively small, rectangular screen, affecting the visibility of nearby AR objects (Fig 1A). Consequently, relatively large downward head orientations were required to get nearby floor-based holographic objects into view [1], see also [20–22]. This might limit the efficacy of AR cueing for alleviating FOG and for assisting and modulating gait [1].

The second generation of Microsoft HoloLens, HoloLens 2, has a larger AR-FOV ($43° \times 29°$ [19]; Fig 1B). The primary objective of this study was to systematically evaluate the effect of AR-FOV size on the head orientation required for viewing and interacting with floor-based real-world and holographic objects. The secondary objective was to evaluate the effect of AR-FOV size on gait speed when interacting with floor-based real-world and holographic objects. We invited a group of healthy, middle-aged adults (40–65 years) to participate in two experiments. In Experiment 1 we compared the head orientation when looking, from a standstill position, at floor-based real-world and holographic objects at various, nearby and far, distances (Fig 2A and 2B, left panels). We expected a greater downward head orientation for looking at nearby holographic compared to nearby real-world objects, but significantly less so for HoloLens 2 than HoloLens 1 given its larger AR-FOV size. In Experiment 2 we compared the head orientations when our participants were interacting with floor-based real-world and holographic 2D and 3D objects onto a 6-meter walkway (Fig 2, all panels). Again, we expected a greater downward head orientation in the vicinity of holographic compared to real-world 2D and 3D objects, but significantly less so for HoloLens 2 than HoloLens 1 given its larger AR-FOV size. Based on previous studies [3, 23], we expected that participants walked slightly

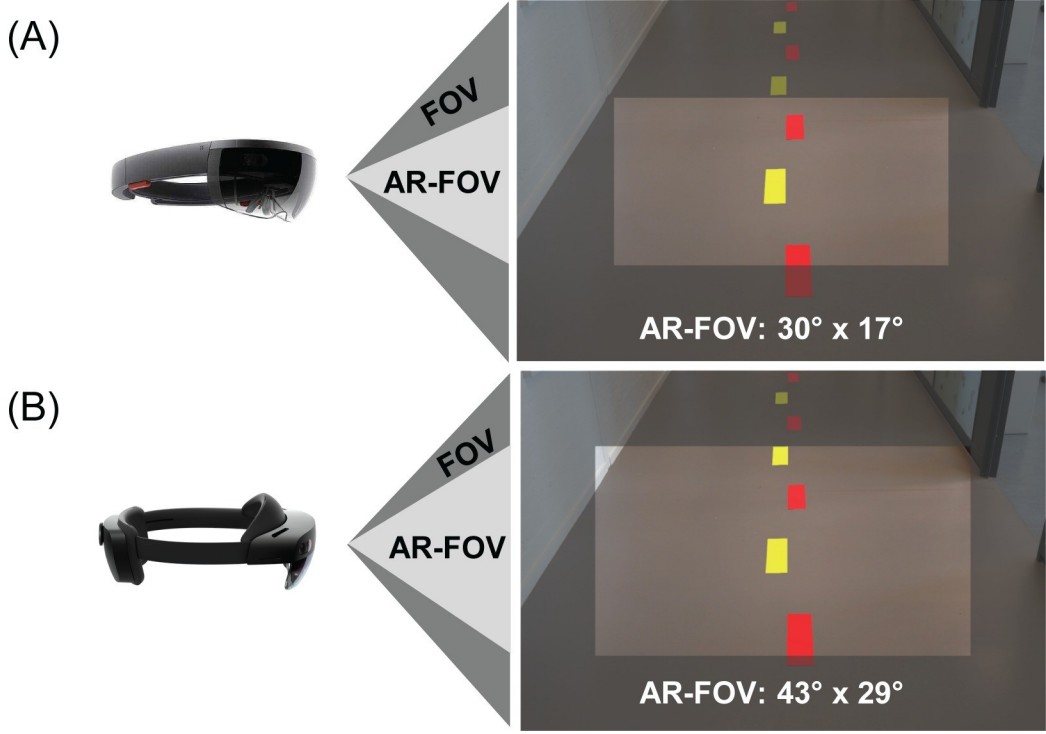

**Fig 1.** Microsoft HoloLens 1 (A) and 2 (B) with a representation of their AR-FOV relative to one's own FOV.

slower when interacting with holographic than with real-world objects, while we had no a-priori expectations on the effect of AR-FOV size.

## Methods

### Participants

A convenience sample of 16 healthy middle-aged adults (mean [range]: 57 [46–65] years), of which 6 females and 10 males, were included in the study. Participants had no conditions interfering with gait function and no restrictions interfering with their depth perception or vision (after potential corrective aids). Participants all provided written informed consent

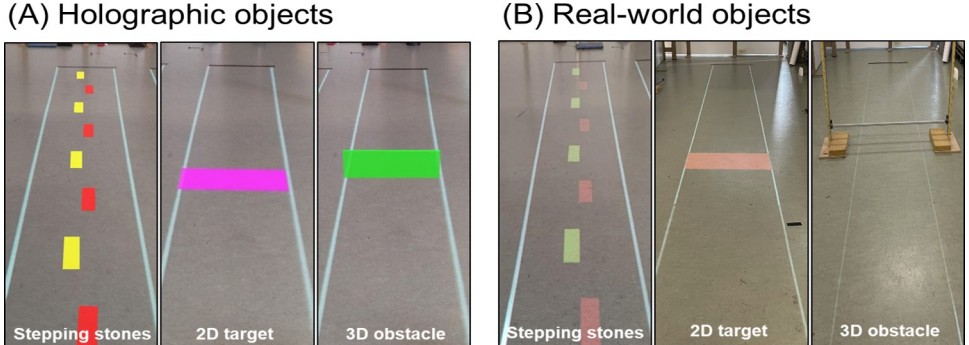

**Fig 2. Experimental setup.** (A) The 6-meter walkways with holographic objects as seen through HoloLens and (B) real-world projected 2D or physical 3D objects on the floor.

**Table 1. Relevant device specifications of HoloLens 1 and 2.**

| | HoloLens 1 | HoloLens 2 |
|---|---|---|
| **Display** | Optical see-through (OST) holographic lenses | Optical see-through (OST) holographic lenses |
| Holographic density | >2.5k radiants (light points per radian) | >2.5k radiants (light points per radian) |
| Holographic resolution | HD 16:9 light engines (1280 × 720) | 2k 3:2 light engines (2048 × 1080) |
| AR-FOV | 30˚ horizontal | 43˚ horizontal |
| | 17˚ vertical | 29˚ vertical |
| **Sensors** | | |
| IMUs | Accelerometer, gyroscope, magnetometer | Accelerometer, gyroscope, magnetometer |
| Sampling frequency | 30 Hz | 30 Hz |
| Depth camera | 1 | 1 |
| Environment understanding camera | 4 | 4 |
| **Human understanding** | | |
| Eye tracking | No | Yes, real-time |
| Hand tracking | Limited to gesture-based finger taps and basic palm moves | Two-handed fully articulated model, direct manipulation |
| **Fit** | | |
| Weight | 579 grams | 566 grams, improved distribution |

before participation. The study was approved by the local Scientific and Ethical Review Board (VCWE-2022-048).

## Experimental design and setup

This n = 16 convenience sample allowed for a full block-randomized counterbalancing of the order of the experimental conditions. Specifically, all experiments comprised two experimental factors with two levels each (hence yielding four possible orders): 1) Type of Object (real-world or holographic) and 2) Type of HoloLens (HoloLens 1 or HoloLens 2) as our manipulation of AR-FOV size (AR-FOV size of HoloLens 2 is greater than that of HoloLens 1; Table 1). Holographic 2D and 3D objects were projected with the HoloLenses (Fig 2A). An overview of the technical specifications of both devices is shown in Table 1. Real-world 2D objects were presented with a projector (EPSON EB-585W, ultra-short-throw 3LCD projector; Fig 2B, left and centered panels) while 3D objects comprised adjustable physical obstacles (Fig 2B, right panel). HoloLens 1 and 2 were used to measure 3D head position and orientation in space, wherein holographic and real-world content was aligned using spatial anchors and headset position calibrated to real-world coordinates using a transformation matrix and predetermined calibration positions. For all experiments, a 6-meter walkway with a width of 75cm was projected, providing a floor-based canvas for presenting 1) 10 stepping stones (10×30cm with an imposed step length of 60cm, where left and right stones were colored yellow and red, respectively; Fig 2A and 2B, left panels), 2) a 2D stepping target (30×75cm, depth×width located midway; Fig 2A and 2B, centered panels) and 3) a 3D obstacle (20×2cm, height×depth, located midway; Fig 2A and 2B, right panels).

## Procedure

Both headset conditions started with calibration of the HoloLens to participant's interpupillary distance using Microsoft's inbuilt calibration software [24] followed by a few minutes of habituation to HoloLens 1 or 2 (depending on the condition) and various holographic objects (i.e., for holographic objects conditions). To methodologically validate if participants perceived differences in the AR-FOV size, we quantified the number of visible real-world and holographic

objects reported by our participants while wearing HoloLens 1 and 2. As expected, participants reported more real-world than holographic objects because the AR-FOV size is smaller than one's own unobstructed FOV size (210˚×150˚ [25]). However, the difference between the reported real-world and holographic objects was significantly smaller for HoloLens 2 because its AR-FOV size is greater than that of HoloLens 1 (Fig 1A and 1B). Participants thus noticed AR-FOV size differences between HoloLenses. Subsequently, the two experiments were sequentially performed in each condition.

**Experiment 1: Head orientations for looking at real-world and holographic objects.** We quantified the head orientation for looking, from a standstill position, at real-world and holographic objects (i.e., stepping stones) onto the floor at various, nearby and far, distances. Participants stood still and looked twice at every stepping stone in a predetermined random order for 8 seconds. The number of the stepping stone which the participant should look at was displayed on a screen next to the participant. Only the yellow stones were included, where the first stone was the one closest to the participant (at 75cm along the walkway), and the fifth stone was the most distant one (at 555cm along the walkway) (Fig 2A and 2B, left panels).

**Experiment 2: Head orientation and gait speed for interacting with real-world and holographic 2D and 3D objects during walking.** We quantified the head orientation and gait speed for interacting with real-world and holographic 2D and 3D objects while walking onto a 6-meter walkway. Participants were instructed to 1) walk as precisely as possible over the sequence of stepping stones at a self-selected comfortable speed (Fig 2A and 2B, left panels), 2) step onto a single 2D stepping target in the center of the walkway (Fig 2A and 2B, centered panels), and 3) cross a 3D obstacle in the center of the walkway (Fig 2A and 2B, right panels). For all three tasks, four trials were performed of which the first was considered a practice trial. At the start and end of each trial, participants fixated straight ahead to a red cross adjusted to their eye level to obtain their baseline head orientation.

## Data and statistical analysis

Headset position and orientation data were recorded with either HoloLens 1 or 2 with a sampling frequency of 30 Hz to analyze participants' head position along the line of progression on the walkway (i.e., increasing along the walkway from the starting position) and head orientation in terms of pitch (i.e., rotation of the headset along the lateral axis such that a downward rotation corresponds to an increase in the orientation angle).

For Experiment 1, the median head orientation from 4-7s for looking at each stepping stone separately was taken and averaged over the two repetitions for each stone (Fig 3A). In case of data irregularities (e.g., high peaks due to sneezing during the trial or external distractions as noted in the experimental logbook), the corresponding episode was excluded from analysis and the median of the remaining episode was included for further statistical analysis. The head orientation for the fifth stepping stone was regarded as the baseline, which was subtracted from the other headset orientations to enable a fair comparison between HoloLens 1 and 2 in case of variations over experimental conditions in the positioning of the AR display on the head relative to the eyes. The resultant headset orientations were subjected to a 2×2×4 (Type of Object × Type of HoloLens × Stone [1 to 4]) repeated-measures ANOVA. The assumption of sphericity was verified according to Girden [26]. If the Greenhouse-Geisser's epsilon exceeded 0.750, the Huynh-Feldt correction was used. Otherwise, the Greenhouse-Geisser correction was applied.

For Experiment 2, baseline head orientation was again subtracted from the head-orientation data for a fair comparison over experimental conditions. Head-orientation data were then expressed as a function of distance along the walkway to enable comparing trials of different

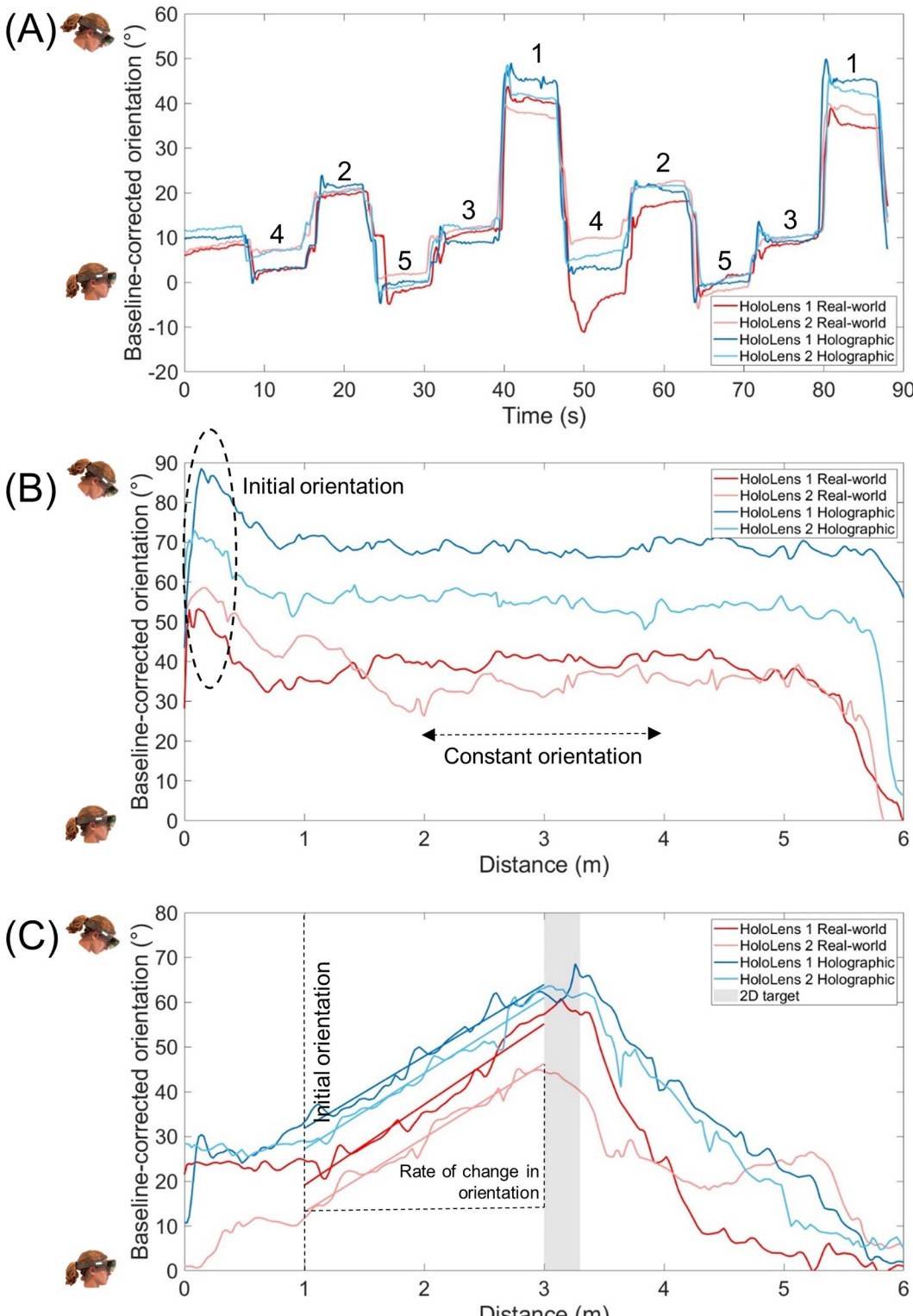

**Fig 3. Representation of head-orientation data for Experiments 1 and 2.** Looking from a stand-still position at real and holographic objects at various, nearby and far, distances (A, Experiment 1; stone numbers increase with distance) and negotiating a continuous sequence of stepping targets (B, Experiment 2, continuous target-stepping task) as well as a single 2D stepping target (C, Experiment 2, 2D target-stepping task). Higher values represent a more downward head orientation. Head orientation was derived from the AR-headset orientation data, normalized to the baseline head orientation.

durations within and between participants. A trial was excluded in case of data irregularities (e.g., due to loss of spatial tracking for 2 out of 768 trials; there were always at least two trials available per condition). The following task-specific outcome parameters were derived, from which the median over repetitions was taken for further statistical analyses:

Continuous target-stepping task:

- The height of the characteristic initial peak, indicative of the *maximal initial downward orientation* of the head to get the stepping stones in view at the start of the trial directly after fixating at the cross at eye level (Fig 3B).

- The median head orientation between 2.0 and 4.0 meters, representing the *constant downward* orientation required for performing the continuous target-stepping task (Fig 3B).

- The median *gait speed* (m/s) between 2.0 and 4.0 meters, calculated from the time required to cover this 2-meter distance.

Incidental 2D target-stepping and 3D obstacle-crossing tasks

- The intercept and slope of the linear fit between the position and orientation of the headset between 1.0 and 3.0 meters along the walkway (Fig 3C), the region where we assumed participants oriented themselves towards the 2D stepping target and 3D obstacle, representing *initial head orientation* and the *rate of change in head orientation*, respectively.

- The median *gait speed* (m/s) for approaching the 2D stepping target or 3D obstacle between 1.0 and 3.0 meters, calculated from the time required to cover this 2-meter distance

Outcome parameters were again subjected to a 2×2 (Type of Object × Type of HoloLens) repeated-measures ANOVA.

Effect sizes are reported as $\eta_p^2$-values. Significant interactions were further explored with paired-samples *t*-tests, focusing -considering the hypotheses- on differences between HoloLenses for both real-world and holographic objects. All data underlying the statistical analyses are available in S1 Table.

## Results

### Experiment 1: Smaller downward orientation for seeing nearby holographic objects with HoloLens 2 than with HoloLens 1

As can be appreciated from Fig 3A, the head orientation varied with object distance in all four conditions, with greater downward orientations for nearby stepping stones, supported by the main effect of Stone ($F(1.55, 23.20) = 1328.67$, $p<0.001$, $\eta_p^2 = 0.989$), with significant orientation differences between all four stones ($p<0.001$). We expected a greater downward head orientation for looking at nearby holographic compared to real-world objects, but less so for HoloLens 2 than HoloLens 1 given its larger AR-FOV size. This was indeed the case, as supported by the three-way interaction ($F(2.07, 30.97) = 14.59$, $p = 0.045$, $\eta_p^2 = 0.191$). Post-hoc paired-samples *t*-tests showed significant head-orientation differences between real-world and holographic objects for most stepping stones, for both HoloLenses alike (Fig 4), notably in magnitude for the two nearest objects. Head orientations for viewing any of the real-world stepping stones did not differ between HoloLenses, while for holographic stepping stones -as expected- a significantly greater downward orientation was required for HoloLens 1 than HoloLens 2, but only for the nearest object (stone 1: $t(15) = 3.377$, $p = 0.004$). AR-FOV size thus mainly affected the orientation for looking at nearby AR objects.

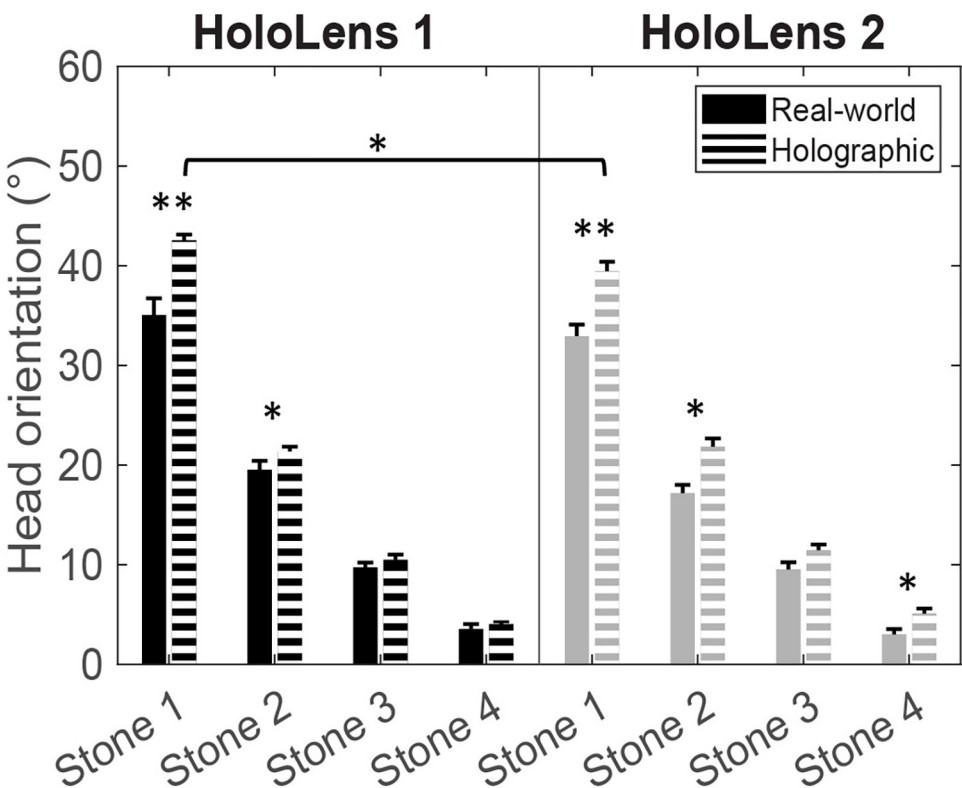

**Fig 4. Data visualization of Experiment 1.** * denotes a significant difference of $p<0.05$ between conditions. ** denotes a significant difference of $p<0.001$ between conditions.

## Experiment 2

**Smaller initial and constant downward orientations for negotiating a sequence of stepping stones with HoloLens 2 than with HoloLens 1.** Head-orientation data showed statistically the same pattern of results over conditions for the continuous target-stepping task. For both parameters (i.e., initial maximal peak and constant orientation), a Type of Object × Type of HoloLens interaction was found ($F(1,15) = 11.33$, $p = 0.004$, $\eta_p^2 = 0.430$ and $F(1,15) = 4.89$, $p = 0.043$, $\eta_p^2 = 0.246$, respectively) on top of main effects for Type of Object (greater downward orientation for holographic than for real-world objects; $F(1,15) = 66.86$, $p<0.001$, $\eta_p^2 = 0.817$ and $F(1,15) = 84.83$, $p<0.001$, $\eta_p^2 = 0.850$, respectively) and Type of HoloLens (greater downward orientation for HoloLens 1 than HoloLens 2 for the initial peak only; $F(1,15) = 24.09$, $p<0.001$, $\eta_p^2 = 0.616$ and $F(1,15) = 0.28$, $p = 0.604$, $\eta_p^2 = 0.018$, respectively). In line with the results for nearby objects of Experiment 1, and as expected given the larger AR-FOV for HoloLens 2 than for HoloLens 1, paired-samples *t*-tests revealed that the initial maximal downward orientation (Fig 5A) as well as the constant downward orientation (Fig 5D) for holographic stepping targets was significantly smaller for HoloLens 2 than for HoloLens 1 ($t(15) = 9.10$, $p<0.001$ and $t(15) = 3.01$, $p = 0.009$, respectively), in the absence of a difference between HoloLenses for real-world stepping targets ($t(15) = 0.63$, $p = 0.535$ and $t(15) = -1.00$, $p = 0.332$, respectively).

**Smaller initial orientation and rate of change in downward orientation for negotiating a single 2D stepping target or 3D obstacle with HoloLens 2 than with HoloLens 1.** In line with our expectations, we found -by and large- a smaller initial orientation (i.e., intercept) and rate of change in downward orientation towards the object (i.e., slope of the linear fit) for

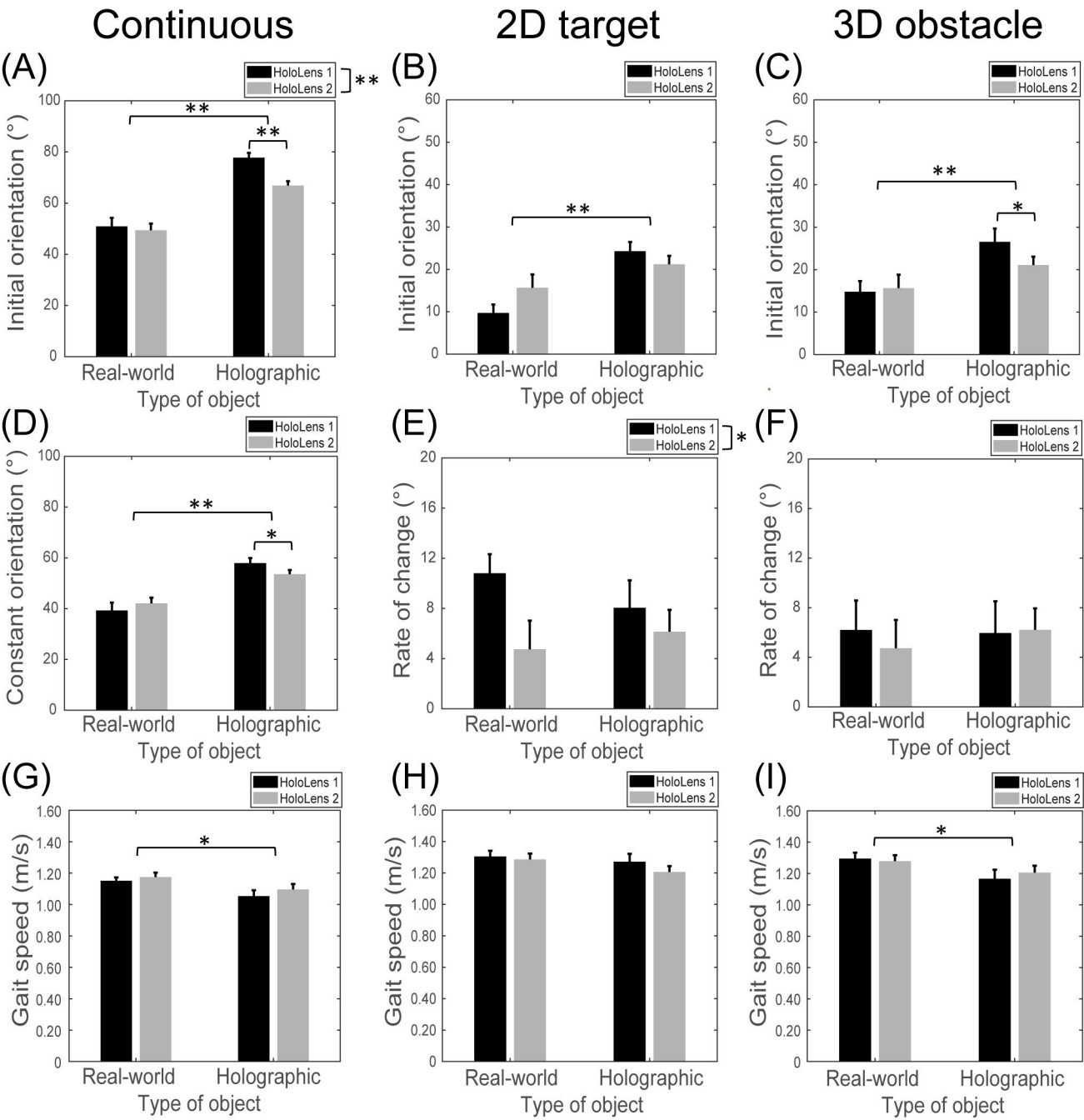

**Fig 5. Data visualization of Experiment 2.** Continuous target-stepping task (A, D, and G), 2D stepping-target task (B, E, and H) and 3D obstacle-crossing task (C, F, and I). * denotes a significant main effect of *p*<0.05 between conditions. ** denotes a significant main effect of *p*<0.001 between conditions.

HoloLens 2 than for HoloLens 1 when negotiating a 2D stepping target and a 3D obstacle. That is, for the initial orientations (Fig 5B and 5C), we found a main effect of Type of Object for both the 2D target-stepping task ($F(1,15) = 49.50$, $p<0.001$, $\eta_p^2 = 0.767$) and the 3D obstacle-crossing task ($F(1,15) = 23.56$, $p<0.001$, $\eta_p^2 = 0.611$), indicating a smaller initial orientation for interacting with real-world than with holographic objects, aligning with the perception

that the AR-FOV is smaller than one's FOV. The main effect of Type of HoloLens was not significant for both tasks (2D stepping target: $F(1,15) = 0.76$, $p = 0.398$, $\eta_p^2 = 0.048$; 3D obstacle crossing: $F(1,15) = 2.64$, $p = 0.125$, $\eta_p^2 = 0.149$), while the Type of Object × Type of HoloLens interaction was borderline significant for 2D target-stepping ($F(1,15) = 4.34$, $p = 0.055$, $\eta_p^2 = 0.225$; Fig 5B) and 3D obstacle-crossing ($F(1,15) = 4.62$, $p = 0.048$, $\eta_p^2 = 0.235$; Fig 4C) tasks, indicating that, at least for holographic 3D obstacle crossing, the initial orientation was significantly smaller for HoloLens 2 than for HoloLens 1 ($t(15) = 2.23$, $p = 0.041$) in the absence of HoloLens differences for real-world 3D obstacles ($t(15) = -0.57$, $p = 0.578$), a finding consistent with our expectations regarding the difference in AR-FOV size between HoloLenses.

For the rate of change in downward orientation (Fig 5E and 5F), only a main effect of Type of HoloLens was observed for the 2D target-stepping task ($F(1,15) = 7.39$, $p = 0.016$, $\eta_p^2 = 0.330$), in the absence of main and interaction effects with the factor Type of Object ($F(1, 15) = 0.55$, $p = 0.469$, $\eta_p^2 = 0.035$ and $F(1,15) = 2.10$, $p = 0.168$, $\eta_p^2 = 0.123$, respectively; Fig 5E) and any effect for the 3D obstacle-crossing task (Type of HoloLens: $F(1,15) = 0.19$, $p = 0.669$, $\eta_p^2 = 0.013$; Type of Object: $F(1,15) = 0.29$, $p = 0.596$, $\eta_p^2 = 0.019$; Interaction: $F(1,15) = 1.17$, $p = 0.296$, $\eta_p^2 = 0.072$; Fig 5F). This implied, consistent with our expectations regarding the difference in AR-FOV size between HoloLenses, a significantly smaller rate of change in the downward orientation when approaching the holographic 2D stepping target for HoloLens 2 than for HoloLens 1.

**Gait speeds are slightly slower for negotiating holographic than real-world objects.** As expected, participants walked slower when negotiating holographic objects compared to real-world objects, as supported by a significant main effect of Type of Object for the continuous target-stepping and the 3D obstacle tasks ($F(1,15) = 12.50$, $p = 0.003$, $\eta_p^2 = 0.455$ and $F(1,15) = 11.34$, $p = 0.004$, $\eta_p^2 = 0.431$, respectively; Fig 5G and 5I) and a borderline significant difference for the 2D stepping-target task ($F(1,15) = 4.48$, $p = 0.051$, $\eta_p^2 = 0.230$; Fig 5H). For all tasks, neither the main effect of Type of HoloLens (Continuous target stepping: $F(1,15) = 1.18$, $p = 0.295$, $\eta_p^2 = 0.073$; 2D stepping target: $F(1,15) = 1.91$, $p = 0.187$, $\eta_p^2 = 0.113$; 3D obstacle: $F(1,15) = 0.26$, $p = 0.621$, $\eta_p^2 = 0.017$) nor the Type of Object × Type of HoloLens interaction (Continuous target stepping: $F(1,15) = 0.18$, $p = 0.674$, $\eta_p^2 = 0.012$; 2D stepping target: $F(1,15) = 1.75$, $p = 0.205$, $\eta_p^2 = 0.105$; 3D obstacle: $F(1,15) = 2.57$, $p = 0.130$, $\eta_p^2 = 0.146$) were significant, indicating that AR-FOV size had no demonstrable effect on gait speed.

## Discussion

The primary objective of this study was to evaluate the effect of AR-FOV size on head orientations for viewing and interacting with floor-based real-world and holographic objects. First, we observed, consistent with our expectations, a smaller downward head orientation when looking at the nearest holographic stone with HoloLens 2 compared to HoloLens 1 (Experiment 1) in the absence of HoloLens differences when looking at the other three, more distant holographic stones, probably because they become smaller and more centered in one's AR-FOV. In Experiment 2, we found that a greater downward head orientation was required when interacting with holographic than real-world 2D or 3D objects during walking, but significantly less so for HoloLens 2 than HoloLens 1 because of its larger AR-FOV size. Results of 2D target-stepping and 3D obstacle-crossing tasks do not match completely: in the 3D holographic obstacle-crossing task the initial downward orientation was significantly smaller for HoloLens 2 than for HoloLens 1, whereas in the 2D target-stepping task the rate of change in downward orientation was significantly smaller for HoloLens 2 than for HoloLens 1. Although in both tasks the net result was a smaller downward head orientation with larger AR-FOV size, the discrepancy over tasks in the effects of the complementary outcome parameters (i.e.,

smaller initial orientation vs. smaller rate of change in downward orientation) probably hints at different gazing strategies when approaching 2D vs. 3D sized objects and/or different gaze-fixation patterns when negotiating targets vs. obstacles [27–29].

For the secondary objective, we evaluated the effect of different AR-FOV sizes on gait speeds when negotiating holographic and real-world objects. Overall, participants walked slightly slower when interacting with holographic objects compared to real-world objects (e.g., a difference of 8.8 cm/s for the continuous target-stepping task representing a 0.14 seconds difference over the calculated 2.0 meters) (Fig 5G–5I). This finding is in line with previous research [3, 23] where participants were suggested to walk more cautiously when approaching holographic objects. No differences were observed between HoloLens 1 and 2, suggesting that the gait speed was not influenced by the AR-FOV size.

This study was conducted in the broader context of AR applications for gait assistance utilizing action-relevant floor-based 2D and 3D objects, like AR cueing for people with neurological conditions like PD [1, 2]. In that regard, two observations are relevant that warrant further investigation, as discussed next.

First, AR cueing may be related to the ecological-psychology concept of affordances introduced by J.J. Gibson to refer to the action possibilities in the environment of an agent [30]. In the context of AR cueing, this environment is partly *mediated* by AR objects yielding functional possibilities for action: 2D AR stepping targets afford to step onto, while a 3D AR obstacle affords to step over. Such affordances exist by the detection of meaningful ecological information from mediated environments. On the one hand, the limited AR display may therefore have limited the availability of meaningful information from AR cues, yielding suboptimal action opportunities, and hence potentially reducing the efficacy of AR cueing as recently discussed [1, 2, 21]. Such reasoning matches with direct empirical proof that a smaller AR-FOV size resulted in an underestimation of gap-stepping affordances [20]. On the other hand, humans exploit a range of behaviors to extend the eye-brain connection by including movements of the eyes, head, and body for revealing and picking up ecological information in the optic array [31]. As E.J. Gibson [32] aptly put it: "*We don't simply see, we look. The visual system is a motor system as well as a sensory one. When we seek information in an optic array, the head turns, the eyes tum to fixate, the lens accommodates to focus, and spectacles may be applied and even adjusted by head position for far or near looking.*" This ecological-psychology notion of perceptual systems may help in interpreting the effects of AR-FOV size on head orientations. That is, downward head orientations are required and appropriate for the pickup of ecological information from (mediated) environments to afford proper foot placement: participants oriented their head more downwards when interacting with nearby objects, for real-world and holographic objects alike, and appropriately changed their head orientation to pick up ecological information in mediated environments, scaled to AR-FOV size. It remains to be seen whether such changes in head orientations are sufficient for proper foot-placement (as afforded by AR stepping stones or the AR 2D stepping target) and obstacle-crossing (as afforded by the AR 3D obstacle) behaviors in mediated environments compared to gait modifications seen with real-world counterparts (but see [5]).

Second, this study included healthy middle-aged adults instead of people with neurological conditions like PD. This might be regarded as a limitation in the broader context of AR cueing because whereas healthy individuals are known to use a feedforward mechanism using exteroceptive information of the environment to plan proactive adjustments of gait [33], people with PD have a stronger dependence on visual information for making online corrections, often using conscious movement processing [34, 35]. Previous studies showed that visual cueing aids the pickup of task-relevant information (including the perception of affordances) in people with PD, improving safety of locomotion and foot-placement accuracy [36–38]. Together,

this indicates that a sufficient AR-FOV size might be even more important for interacting with floor-based holographic objects for people with PD than for healthy individuals. Furthermore, a limited AR-FOV size could promote an even more stooped posture in people with PD, which is associated with poorer balance and an increased risk of falls, emphasizing the importance of a sufficient AR-FOV size even more [39, 40].

Another limitation in the broader context of AR cueing is that we only examined the effect of AR-FOV size on head orientations and performed gait speed, without examining possible effects on gait modulation in terms of step length, cadence and obstacle-crossing heights and lengths, which are relevant aspects for effective AR cueing. Future research with people with PD is warranted to investigate if an increased AR-FOV size improves the efficacy of AR cues for alleviating FOG and for assisting and modulating gait. Such studies may benefit from including or comparing HoloLens 2 with other AR devices with an even larger AR-FOV size, like Magic Leap 2 (45˚×55˚ [41]) and the recently announced Meta Orion (with an AR-FOV expected to be even larger). Alternatively, one could consider including a video see-though device (e.g., Apple vision Pro) instead of only optical see-through devices as used in the current study, allowing for more fine grained and wider evaluations of the effect of AR-FOV size on head orientation and gait speed for interacting with nearby AR content. Video see-through devices, however, seem less suitable for assisting and modulating gait in health-related fields, mainly because of the risk of losing the vision of the real environment in case of system errors or an empty battery, which could lead to dangerous situations in daily life as wearers then find themselves in a pitch-black environment. Finally, studies may benefit from including eye-tracker data, present in state-of-the-art AR headsets, to confirm the above-mentioned different gaze strategies associated with negotiating targets vs. obstacles and/or 2D vs. 3D objects in real-world and AR environments.

To summarize, the downward head orientation was reduced for viewing and interacting with holographic objects along the line of sight with HoloLens 2 compared to HoloLens 1, particularly for objects nearby. Still, interacting with holographic objects required a somewhat greater downward head orientation compared to interacting with real-world objects, which may or may not affect affordance-based actions. Gait speed was not affected by AR-FOV size but was slightly but significantly slower for negotiating holographic objects than real-world objects. The ongoing technology development of AR devices will at least bring AR-FOV size closer to one's FOV, from which AR applications for modulating and assisting gait in neurological conditions like PD will probably benefit.

## Supporting information

**S1 Table. Data file.**
(XLSX)

## Acknowledgments

We would like to thank all participants for their voluntary participation.

## Author Contributions

**Conceptualization:** Eva M. Hoogendoorn, Daphne J. Geerse, Melvyn Roerdink.

**Formal analysis:** Eva M. Hoogendoorn, John F. Stins.

**Funding acquisition:** Daphne J. Geerse, Melvyn Roerdink.

**Investigation:** Eva M. Hoogendoorn, Jip Helsloot.

**Methodology:** Eva M. Hoogendoorn.

**Software:** Bert Coolen.

**Supervision:** Daphne J. Geerse, Melvyn Roerdink.

**Visualization:** Eva M. Hoogendoorn.

**Writing – original draft:** Eva M. Hoogendoorn.

**Writing – review & editing:** Daphne J. Geerse, Jip Helsloot, Bert Coolen, John F. Stins, Melvyn Roerdink.

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
