## [Decision Letter · Decision Letter 0]

18 Jan 2024

PONE-D-23-31697A greater mixed-reality field of view improves interaction with nearby holographic objectsPLOS ONE

Dear Dr. Hoogendoorn,

Thank you for submitting your manuscript to PLOS ONE. After careful consideration, we feel that it has merit but does not fully meet PLOS ONE’s publication criteria as it currently stands. Therefore, we invite you to submit a revised version of the manuscript that addresses the points raised during the review process.

Considering the comments received by the reviewers (i.e., two Major Revisions) I recommend considering the manuscript after a careful revision of the authors. Please find below a summary of the reviewers' comments and detailed indications to improve the quality of the manuscript. 

We look forward to receiving your revised manuscript.

Kind regards,

Alberto Cannavò

Academic Editor

PLOS ONE

Journal Requirements:

**Additional Editor Comments:**

The first reviewer provides an assessment of the study, warning about the minimal difference in fields of view (FOVs) and emphasizing the fact that having two conditions is limiting. Moreover, the review claims for clearer objectives and research question formulation.

Specific inquiries are made regarding the experiments and the importance of statistical power analysis is highlighted too. Additionally, the reviewer encourages a detailed analysis of the differences between Hololens 1 and Hololens 2

The second reviewer raises major concerns about the study's statistical reliability due to the small sample size and questions the novelty of the paper. Minor issues such as language correction and figure quality are also noted.

Moving from these considerations and due to the number of comments, the paper could be considered for publication after a major revision.

Reviewers' comments:

Reviewer's Responses to Questions

**Comments to the Author**

1. Is the manuscript technically sound, and do the data support the conclusions?

Reviewer #1: Partly

Reviewer #2: Yes

2. Has the statistical analysis been performed appropriately and rigorously? 

Reviewer #1: Yes

Reviewer #2: No

3. Have the authors made all data underlying the findings in their manuscript fully available?

Reviewer #1: Yes

Reviewer #2: Yes

4. Is the manuscript presented in an intelligible fashion and written in standard English?

Reviewer #1: Yes

Reviewer #2: Yes

5. Review Comments to the Author

Reviewer #1: General

-The difference in FOVs in this study is small. The reveiwer also consirers that only two conditions are really limited.

Introduction

-There are OST-HMD and VST-HMD. The auhotrs had better mention more conclusive MR-based gait modules. (but optional, not necessary)

-Citing and discussing depth perception study might be helpful.

-The objectives should be mentioned more cleary.

-How to make the research question is unclear. The reviewr considers that it is too natural that higher FOV obviously leads to in the higher number of obstalces that people can recognise. What elements are unknown and how to investigate them should be explained more clearly.

Experiment 1: visibility of real-world and holographic objects

-How was depth perception of the participants.

Experiment 2: head orientations for looking at real-world and holographic objects

-How to quantify the head orientation?

-Why was the closest condition 75cm along the walkway?

Experiment 3: head orientations for interacting with real-world and holographic 2D and 3D objects during walking

-How to quantify the head orientation?

-Walking velocity should be reported because it affects a human vision, such as optical flow, and preparation time for stepping obstacles.

-Results

Please add statistical power.

Discussion

-In this study, the OST-HMD was used for investigation. However, when it comes to investigation of the FOV effects, VST-HMD is useful to make the conditions of the FOV. The reviewer hopes to see insights of meaning of using OST-HMD for investigation of the FOV effects.

-Some significant differences were found between Hololens 1 and Hololens 2. But how this differences affect human motion is unclear. The reviewer wants to encourage this analysis (literature based discussion is possible).

-The effect of downward head orientation on human cognition also might be able to be discussed.

Reviewer #2: The Authors analyzed the influence of the Mixed Reality (MR) field of view in the case of users suffering from Parkinson’s disease. The head orientations required for viewing and interacting with real-world and holographic floor-based objects during standstill and walking conditions were studied.

Major Comments

From a statistical point of view, the study sample is too small to draw any reliable conclusions. Additionally, some of the observations from the experiments are derived from the technical specifications of the hardware (differences between HoloLens 1 and HoloLens 2). In the context of device specifications, it is not clear what the new paper brings.

Minor Comments

The English language requires correction.

The included figures are of very low quality.

The phrase „Mixed” Reality should be written in capital letters.

To summarize, the proposed paper suffers significant imperfections. I do not recommend the paper for publication in the following form. At this stage, I recommend at least very major revision.

6. PLOS authors have the option to publish the peer review history of their article (what does this mean?). If published, this will include your full peer review and any attached files.

Reviewer #1: No

Reviewer #2: No

---

## [Author Response · Author response to Decision Letter 0]

26 Feb 2024

Dear professor Cannavò,

Thank you very much for your interest in our manuscript titled "A greater augmented-reality field of view improves interaction with nearby holographic objects." We appreciate the thoughtful comments of the reviewers and we have revised our manuscript accordingly. In brief, we have included an extra analysis of the effect of differences in AR field of view on gait speed and we have added a detailed overview with specifications of HoloLens 1 and 2. Changes have been highlighted in the copy of the revised manuscript and are also specified below. As requested, we have also uploaded a clean copy. 

We hope that our manuscript is now suitable for publication in PLOS ONE. 

Sincerely, on behalf of my co-authors, 

Eva Hoogendoorn

Our point-by-point response:

Reviewer 1

1. The difference in FOVs in this study is small. The reviewer also considers that only two conditions are really limited.

Response: We do acknowledge that it would be interesting to include more conditions with larger differences in AR-FOV. To do so, VST-HMDs are required. Our study, however, focused on OST-HMDs, see also our response on comment 2 of reviewer 1. We included this relevant comment as a recommendation for future research in the Discussion section:

“Alternatively, one could consider including a video see-though device (e.g., Apple vision Pro) instead of only optical see-through devices used in the current study, allowing for a more fine grained and wider evaluations of the effect of AR-FOV size on head orientation and gait speed for interacting with nearby AR content.”

2. There are OST-HMD and VST-HMD. The authors had better mention more conclusive MR-based gait modules. (but optional, not necessary).

Response: Thank you for this relevant comment. Our research only included OST-HMDs. Based on the reviewers' comment, we expressed this more clearly in the revised manuscript, see Table 1 in the Methods section for a detailed overview of specifications of the HMDs. We are particularly interested in OST-HMDs, because of their potential for gait assistance, gait modulation, gait-and-balance assessment and gait-and-balance training (as was already described in the Introduction). We do agree that VST-HMDs are interesting to include in future research and we adjusted the text accordingly as specified in our response to your comment 1. 

3. Citing and discussing depth perception study might be helpful.

Response: The reviewer rightly mentioned the importance of depth perception. Since we added problems with depth perception in the exclusion criteria (see also our response to your comment 6), a discussion on depth perception was not included. 

4. The objectives should be mentioned more clearly.

Response: We have revisited our objectives, amended them in light of the reviewers’ comments to examine the effect of AR-FOV on gait characteristics (see also our response to your comment 9 below) and adjusted them for the sake of clarity. The objectives now read as follows (see Introduction section): 

“The primary objective of this study was to systematically evaluate the effect of AR-FOV size on the head orientation required for viewing and interacting with floor-based real-world and holographic objects. The secondary objective was to evaluate the effect of AR-FOV size on gait speed when interacting with floor-based real-world and holographic objects.”

5. How to make the research question is unclear. The review considers that it is too natural that higher FOV obviously leads to the higher number of obstacles that people can recognize. What elements are unknown and how to investigate them should be explained more clearly.

Response: We agree with the reviewer that it is rather obvious that a larger AR-FOV size leads to a higher number of visible AR objects. Therefore, we decided to move this finding (original Experiment 1) to the Methods section and presented this as a methodological validation of hardware specifications (see also our response to comment 2 of reviewer 2): 

“To methodologically validate if participants perceived differences in the AR-FOV size, we quantified the number of visible real-world and holographic objects reported by our participants while wearing HoloLens 1 and 2. As expected, participants reported more real-world than holographic objects because the AR-FOV size is smaller than one’s own unobstructed FOV size (210º×150º [25]). However, the difference between the reported real-world and holographic objects was significantly smaller for HoloLens 2 because its AR-FOV size is greater than that of HoloLens 1 (Figs 1A-B).”

However, it still remains unclear what the effect of the AR-FOV size is on human perceptual-motor behavior in terms of head orientation required to interact with the presented AR content as well as the resultant gait speed during these interactions, as evaluated in this article. We therefore specified this as our objectives (see also our response to your comment 4 above). 

6. How was depth perception of the participants.

Response: The reviewer rightly hinted on the importance of depth perception of the participants. Two persons that had problems with depth perception were excluded from participating. We specified this as follows in the Methods section of this revision: “Participants had no conditions interfering with gait function and no restrictions interfering with their depth perception or vision (after potential corrective aids).” 

7. How to quantify the head orientation?

Response: Head orientation was derived from the AR-headset orientation data, normalized to baseline orientation (i.e., to correct for potential between-headset-condition differences in positioning of the AR visor in front of participants’ eyes), as described in detail in the Methods section. We have expanded the explanation to improve visibility and clarity in the text: 

“Headset position and orientation data were recorded with either HoloLens 1 or 2 with a sampling frequency of 30 Hz to analyze participants’ head position along the line of progression on the walkway (i.e., increasing along the walkway from the starting position) and head orientation in terms of pitch (i.e., rotation of the headset along the lateral axis such that a downward rotation corresponds to an increase in the orientation angle).”

We also added a short explanation within the description of figure 3: “Head orientation was derived from the AR-headset orientation data, normalized to the baseline head orientation.”

8. Why was the closest condition 75cm along the walkway?

Response: this was a somewhat arbitrary choice, thought to resemble a natural initiation step length of 60cm (i.e., 75 cm refers to the center of the 30cm stepping target). 

9. Walking velocity should be reported because it affects a human vision, such as optical flow, and preparation time for stepping obstacles.

Response: Adding gait speed is an excellent suggestion, as indeed this could have been affected by the AR-FOV size and associated changes in head orientation required to interact with AR content. We could derive gait speeds from the AR positional headset data. We have included gait speed in the revised manuscript as a secondary objective of the study. The effects, if any, were small in magnitude but significant for some conditions, with slightly slower gait speeds (~8.8 cm/s) for interacting with floor-based holographic than real-world content.

10. Please add statistical power.

Response: From a statistical point of view, it is unusual to add a power-analysis post-hoc. However, we do report the effect size (as partial eta squared) so that future studies are in a better position to perform an a-priori power analysis. 

11. In this study, the OST-HMD was used for investigation. However, when it comes to investigation of the FOV effects, VST-HMD is useful to make the conditions of the FOV. The reviewer hopes to see insights of meaning of using OST-HMD for investigation of the FOV effects.

Response: Thank you for this relevant comment. We are particularly interested in OST-HMDs, because of its potential for gait assistance, gait modulation, gait-and-balance assessment and gait-and-balance training in real-world settings (as described in the Introduction, see also our response to your comment 2). With OST-HDMs, system errors or empty battery impact the AR content, but not the visibility of the real world. This is, however, different with VST-HMDs. We therefore focused on OST-HMDs that are available today. We included an explanation in the Discussion section:

“ Alternatively, one could consider including a video see-though device (e.g., Apple vision Pro) instead of only optical see-through devices as used in the current study, allowing for more fine grained and wider evaluations of the effect of AR-FOV size on head orientation and gait speed for interacting with nearby AR content. Video see-through devices, however, seem less suitable for assisting and modulating gait in health-related fields, mainly because of the risk of losing the vision of the real environment in case of system errors or an empty battery, which could lead to dangerous situations in daily life as wearers then find themselves in a pitch-black environment.”

Note: since recent studies with OST-HMDs more often refer to the technology as augmented-reality instead of mixed-reality, we decided to change the term ‘mixed-reality (MR)’ into ‘augmented-reality (AR)’ throughout the manuscript.

12. Some significant differences were found between Hololens 1 and Hololens 2. But how this differences affect human motion is unclear. The reviewer wants to encourage this analysis (literature based discussion is possible).

Response: Thank you for this excellent suggestion, which we addressed by including gait speed as a secondary objective, as detailed in our response to your comment 9. 

13. The effect of downward head orientation on human cognition also might be able to be discussed.

Response: This is an interesting comment, but it lies outside the scope of this study.

Reviewer 2

1. The Authors analyzed the influence of the Mixed Reality (MR) field of view in the case of users suffering from Parkinson’s disease. The head orientations required for viewing and interacting with real-world and holographic floor-based objects during standstill and walking conditions were studied. From a statistical point of view, the study sample is too small to draw any reliable conclusions. 

Response: Our study sample size of 16 participants is a convenience sample, which allowed for a fully block-randomized counterbalancing of the order of the experimental conditions, suitably large for a repeated-measures design wherein each participant performs all conditions. 

We would like to emphasize that our study is a technical validation study executed with healthy adults (and not with people with Parkinson’s disease like reviewer 2 suggested). It is not unusual to include 10-20 participants in technical validation studies as has been done in relevant previous studies [1-3], to name a few. Furthermore, we implemented a repeated-measure design and executed all walking trials three times, to increase the reliability of our results. We are therefore confident that our sample size of n=16 was sufficient for this study, as corroborated by profound effects with large effects sizes in line with our expectations. 

2. Some of the observations from the experiments are derived from the technical specifications of the hardware (differences between HoloLens 1 and HoloLens 2). In the context of device specifications, it is not clear what the new paper brings.

Response: We agree with the reviewer that the observations from Experiment 1 logically follow from differences in technical specifications between the HoloLenses. As detailed in our response to the related comment 5 of Reviewer 1, we decided to move this finding (original Experiment 1) to the Methods section and presented this as a methodological validation of hardware specifications, which are now also detailed for HoloLens 1 and 2 in the new Table 1. 

“To methodologically validate if participants perceived differences in the AR-FOV size, we quantified the number of visible real-world and holographic objects reported by our participants while wearing HoloLens 1 and 2. As expected, participants reported more real-world than holographic objects because the AR-FOV size is smaller than one’s own unobstructed FOV size (210º×150º [25]). However, the difference between the reported real-world and holographic objects was significantly smaller for HoloLens 2 because its AR-FOV size is greater than that of HoloLens 1 (Figs 1A-B).”

However, it still remains unclear what the effect of the AR-FOV size is on human perceptual-motor behavior in terms of head orientation required to interact with the presented AR content as well as the resultant gait speed during these interactions, as evaluated in this article. 

3. The English language requires correction.

Response: Point well taken. We made several textual corrections throughout the manuscript.

4. The included figures are of very low quality.

Response: It remains unclear what the reviewer meant with very low quality. The figure files are uploaded in the Preflight Analysis and Conversion Engine (PACE) digital diagnostic tool to ensure that figures meet PLOS requirements. 

References

1. Binaee K, Diaz GJ. Assessment of an augmented reality apparatus for the study of visually guided walking and obstacle crossing. Behavior Research Methods. 2019;51(2):523-31. doi: 10.3758/s13428-018-1105-9.

2. Chan ZYS, MacPhail AJC, Au IPH, Zhang JH, Lam BMF, Ferber R, Cheung RTH. Walking with head-mounted virtual and augmented reality devices: Effects on position control and gait biomechanics. PLOS ONE. 2019;14(12):e0225972. doi: 10.1371/journal.pone.0225972.

3. Koop MM, Rosenfeldt AB, Johnston JD, Streicher MC, Qu J, Alberts JL. The HoloLens Augmented Reality System Provides Valid Measures of Gait Performance in Healthy Adults. IEEE Transactions on Human-Machine Systems. 2020;50(6):584-92. doi: 10.1109/THMS.2020.3016082.

---

## [Decision Letter · Decision Letter 1]

7 May 2024

PONE-D-23-31697R1

A larger augmented-reality field of view improves interaction with nearby holographic objects

PLOS ONE

Dear Dr. Hoogendoorn,

Thank you for submitting your manuscript to PLOS ONE. After careful consideration, we have decided that your manuscript does not meet our criteria for publication and must therefore be rejected.

Specifically:

**ACADEMIC EDITOR: **

Unfortunately, one of the reviewers has raised critical concerns regarding crucial aspects of the manuscript, i.e., the technical contribution, rigorous of the definitions, and the experiments. For these reasons I recommend rejecting the manuscript.

I am sorry that we cannot be more positive on this occasion, but hope that you appreciate the reasons for this decision.

Kind regards,

Alberto Cannavò

Academic Editor

PLOS ONE

**Additional Editor Comments:**

Although one of the reviewers has considered the concerns raised in the first round addressed, the second one was not positive about the new version of the manuscript.

More specifically, he remarked on limitations in the technical contribution, lamented a non-representative sample size, and asked for several changes in the text. Moving from these considerations the authors are recommended to revise and resubmit the manuscript.

Reviewers' comments:

Reviewer's Responses to Questions

**Comments to the Author**

1. If the authors have adequately addressed your comments raised in a previous round of review and you feel that this manuscript is now acceptable for publication, you may indicate that here to bypass the “Comments to the Author” section, enter your conflict of interest statement in the “Confidential to Editor” section, and submit your "Accept" recommendation.

Reviewer #1: All comments have been addressed

Reviewer #2: All comments have been addressed

2. Is the manuscript technically sound, and do the data support the conclusions?

Reviewer #1: Yes

Reviewer #2: No

3. Has the statistical analysis been performed appropriately and rigorously? 

Reviewer #1: Yes

Reviewer #2: No

4. Have the authors made all data underlying the findings in their manuscript fully available?

Reviewer #1: Yes

Reviewer #2: Yes

5. Is the manuscript presented in an intelligible fashion and written in standard English?

Reviewer #1: Yes

Reviewer #2: Yes

6. Review Comments to the Author

Reviewer #1: Thank you for your modofication.

I have checked the all answerers and modifications.

All my comments have been addressed.

Reviewer #2: The Authors do not analyze Parkinson's disease, so they should remove references or any mention to it in the abstract and introduction.

The Authors analyze/perform experiments in the area of technical properties/features of the Mixed Reality equipment itself. The added value of the paper is not clear (the obtained results are an obvious consequence of the technical parameters of the HoloLens generation).

Conducting statistical research on 16 cases divided into groups (10 individuals versus 6 individuals) does not provide reliable conclusions.

The title concerns Augmented Reality as well as main text and the Authors investigate Mixed Reality. Which indicates their ignorance of the nomenclature or specifics of the equipment.

7. PLOS authors have the option to publish the peer review history of their article (what does this mean?). If published, this will include your full peer review and any attached files.

Reviewer #1: No

Reviewer #2: No

- - - - -

---

## [Author Response · Author response to Decision Letter 1]

19 Jun 2024

Dear professor Cannavò,

Thank you for your editorial feedback and for the time you have taken to review our manuscript. We appreciate the positive feedback from Reviewer 1 who recommended to accept our manuscript for publication in PLOS ONE. 

We also fully understand that you have to take Reviewer 2's concerns seriously. However, we respectfully disagree with Reviewer 2's assessment. He or she clearly misread (or even did not fully read) our manuscript: there was no between-groups comparison in our manuscript, contrary to what was suggested by the reviewer. Our study is a technical validation study conducted with 16 healthy adults (not persons with Parkinson’s disease as again was misread by the same Reviewer in the initial round of reviews). All participants performed all headset [HoloLens 1, HoloLens 2] by content [real-world cues, AR cues] conditions in a fully counter-balanced within-subject repeated-measures design, which was also addressed in our previous response letter to a similar comment of Reviewer 2. Reviewer 2 further mentioned that there was an "ignorance of the nomenclature or specifics", which we believe is not accurate, as explained in more detail in the point-by-point response below. In brief, Mixed Reality (MR, where the Reviewer is clearly a fan of) is the umbrella term used for all realities within Milgram’s famous virtuality spectrum ranging from Reality to Virtual Reality, where closer to Reality the term Augmented Reality (AR, which we used in our manuscript) is pertinent while closer to Virtual Reality (VR) the term Augmented Virtuality (AV) is pertinent. Hence, both the use of AR and MR terminology would be accurate for our manuscript, with AR being more specific than MR in that regard as we augment the real world with holographic cues (AR, closer to the Reality than to the Virtual Reality endpoint of Milgram’s virtuality spectrum). 

Besides that, Reviewer 2 stated that the obtained results are an obvious consequence of the different technical parameters of HoloLens 1 and 2, thereby implicitly evaluating our manuscript on perceived significance. We believe that this interpretation reflects a personal preference (similar to the MR vs. AR choice), not meeting the criteria set forth by PLOS ONE. As PLOS ONE evaluates research based on scientific validity, robust methodology, and high ethical standards—rather than perceived significance—we aimed to provide a comprehensive and methodologically sound evaluation of how differences in technical parameters between the two HoloLens generations affected perception of holographic content as well as the interaction with them in terms of gait speed and the required head orientations to pick up task-relevant AR and real-world information. Our study offers detailed insights in those regards that are not immediately apparent without scientific investigation. Therefore, we believe that our research meets the criteria for publication (Journal Information | PLOS ONE). 

Given these points, we kindly request to respectfully reweigh Reviewer 2’s concerns (which are factual non-issues given the publication criteria of PLOS ONE) and to reconsider our revised manuscript for publication in PLOS ONE. 

Sincerely, on behalf of my co-authors, 

Eva Hoogendoorn

Our point-by-point response:

Reviewer 1

I have checked the all answerers and modifications.

All my comments have been addressed.

Thank you very much for reviewing our manuscript and endorsing it for publication. 

Reviewer 2

1. The Authors do not analyze Parkinson's disease, so they should remove references or any mention to it in the abstract and introduction.

Response: Thank you for your feedback. We have removed the reference to Parkinson's disease from the abstract. However, we have retained the reference to Parkinson’s disease in the introduction to provide essential context regarding the broader application of AR technology in general, and the significance of AR cueing applications in specific.

2. The Authors analyze/perform experiments in the area of technical properties/features of the Mixed Reality equipment itself. The added value of the paper is not clear (the obtained results are an obvious consequence of the technical parameters of the HoloLens generation).

Response: While we understand and respect your perspective, we believe this interpretation of (presumably) the difference in the AR/MR field of view size between the two HoloLens generations may be more reflective of your perceived significance of our work than being ground for rejection of our paper given the publications criteria set forth by PLOS ONE.

That is, the journal's focus on "quality-based selection" emphasizes the importance of scientific validity, robust methodology, and high ethical standards over perceived significance. Our research is designed to adhere to these principles. Specifically, we aimed to provide a comprehensive and methodologically sound evaluation of how differences in technical parameters (FOV-size) between the two HoloLens generations affected the perception and pick-up of holographic information (in comparison to real-world information) as well as the interaction with this information in terms of gait speed and the required head orientations to pick up task-relevant AR and real-world information, offering insights that are not immediately apparent without scientific investigation. Whether this is perceived as a significant contribution to science or not is at best a subjective interpretation, but not a PLOS ONE publication criterium. 

3. Conducting statistical research on 16 cases divided into groups (10 individuals versus 6 individuals) does not provide reliable conclusions.

Response: We are afraid that you misread our study: there was no division of groups. Our study is a technical validation study conducted with 16 healthy adults, all performing all four headset (HoloLens 1 and 2) by content (real-world and holographic) conditions in a fully counter-balanced within-subject repeated-measures design. 

We would like to point out that this was already addressed in our previous response letter: 

“We would like to emphasize that our study is a technical validation study executed with healthy adults (and not with people with Parkinson’s disease like reviewer 2 suggested). It is not unusual to include 10-20 participants in technical validation studies as has been done in relevant previous studies [1-3], to name a few. Furthermore, we implemented a repeated-measures design and executed all walking trials three times, to increase the reliability of our results. We are therefore confident that our sample size of n=16 was sufficient for this study, as corroborated by profound effects, in line with our expectations, with large effect sizes.”

4. The title concerns Augmented Reality as well as main text and the Authors investigate Mixed Reality. Which indicates their ignorance of the nomenclature or specifics of the equipment.

Response: We feel that this comment is not only offensive but also inaccurate. As frontrunners in the field, we are well aware of nomenclature and of the importance of its precise use in light of Milgram’s virtuality continuum, ranging from Reality (real environment) on the one end to Virtual Reality (virtual environment) on the other end of the continuum. Within these extremes, we find ourselves in a mixed reality (MR), a rather broad and unspecific umbrella term covering all realities except pure Reality and pure Virtual Reality. To be more precise, mixed-realities that are closer to the Reality end than to the Virtual Reality end of the continuum have been coined Augmented Reality (AR) while mixed-realities closer to the Virtual Reality than to the Reality end have been coined Augmented Virtuality. Clearly, our study, in which we augmented the real world with holographic cues, represents Augmented Reality. 

Milgram, P., and Kishino, F. (1994). A taxonomy of mixed reality visual displays. IEICE Trans. Inform. Syst. 77, 1321–1329.

---

## [Editor Report · Decision Letter 2]

26 Sep 2024

A larger augmented-reality field of view improves interaction with nearby holographic objects

PONE-D-23-31697R2

Dear Dr. Hoogendoorn,

We’re pleased to inform you that your manuscript has been judged scientifically suitable for publication and will be formally accepted for publication once it meets all outstanding technical requirements.

Kind regards,

Omnia Hamdy, PhD

Academic Editor

PLOS ONE
---

## [Editor Report · Acceptance letter]

10 Oct 2024

PONE-D-23-31697R2 

PLOS ONE

Dear Dr. Hoogendoorn, 

I'm pleased to inform you that your manuscript has been deemed suitable for publication in PLOS ONE. Congratulations! Your manuscript is now being handed over to our production team.

Kind regards, 

on behalf of

Dr. Omnia Hamdy 

Academic Editor

PLOS ONE